# Polysaccharides Release in a Laboratory-Scale Batch Hydrothermal Pretreatment of Wheat Straw under Rigorous Isothermal Operation

**DOI:** 10.3390/molecules27010026

**Published:** 2021-12-22

**Authors:** Felicia Rodríguez, Efrén Aguilar-Garnica, Adrián Santiago-Toribio, Arturo Sánchez

**Affiliations:** 1Laboratorio de Futuros en Bioenergía, Centro de Investigación y de Estudios Avanzados del IPN (CINVESTAV) Unidad Guadalajara, Av. del Bosque 1145, Col. El Bajío, Zapopan 45019, JA, Mexico; felicia.rodriguez@edu.uag.mx (F.R.); adrian.santiago.t@gmail.com (A.S.-T.); 2Departamento de Ciencias Biotecnológicas y Ambientales, Universidad Autónoma de Guadalajara, Av. Patria 1201, Col. Lomas del Valle, Zapopan 45129, JA, Mexico

**Keywords:** hydrothermal pretreatment, wheat straw, lignocellulosic biomass, polysaccharides release

## Abstract

Hydrothermal pretreatment (HP) is an eco-friendly process for deconstructing lignocellulosic biomass (LCB) that plays a key role in ensuring the profitability of producing biofuels or bioproducts in a biorefinery. At the laboratory scale, HP is usually carried out under non-isothermal regimes with poor temperature control. In contrast, HP is usually carried out under isothermal conditions at the commercial scale. Consequently, significant discrepancies in the values of polysaccharide releases are found in the literature. Therefore, laboratory-scale HP data are not trustworthy if scale-up or retrofitting of HP at larger scales is required. This contribution presents the results of laboratory-scale batch HP for wheat straw in terms of xylan and glucan release that were obtained with rigorous temperature control under isothermal conditions during the reaction stage. The heating and cooling stages were carried out with fast rates (43 and −40 °C/min, respectively), minimizing non-isothermal reaction periods. Therefore, the polysaccharide release results can be associated exclusively with the isothermic reaction stage and can be considered as a reliable source of information for HP at commercial scales. The highest amount of xylan release was 4.8 g/L or 43% obtained at 180 °C and 20 min, while the glucan release exhibited a maximum of 1.2 g/L or 5.5%. at 160 °C/180 °C and 30 min.

## 1. Introduction

Produced at a rate of 980 million tons per year, wheat straw (WS) is currently one of the most abundant agricultural residues worldwide [1]. This large availability, along with its high polysaccharides content (glucan and xylan mainly), makes WS a suitable lignocellulosic feedstock to produce bioethanol and other valuable products in biorefineries [2]. Pretreatment is the first stage in a biochemical platform biorefinery aimed at the extraction of soluble polysaccharides and the deconstruction of the lignocellulosic biomass (LCB) matrix [3].

A wide variety of pretreatment technologies have been proposed in the literature. Among them, hydrothermal pretreatment (HP) is an attractive option that can handle moisture-rich LCB with low energy requirements, making it suitable for large-scale applications [4,5]. The chemical reaction mechanism of HP uses the hydronium ions generated in situ by liquid hot water autoionization as a catalyst [6]. Therefore, HP is considered to be an environmentally friendly pretreatment. Batch-mode HP is carried out in three sequential stages: heating, reaction, and cooling. At the reaction stage, HP operates under saturated steam conditions at mid-range temperatures (120–230 °C) and reaction times between 0 and 3 h [7]. Below 220 °C, xylan hydrolysis is favored over glucan hydrolysis [8]. Consequently, the solid fraction of the pretreated biomass is enriched in cellulose whereas the liquid fraction is mainly composed of xylose and xylo-oligomers (XOOs). Usually, XOOs with a polymerization degree ranging from 2 to 12 units [9] are considered as feedstock for several commercial products in the cosmetic, food, and health industries [10]. Although xylan release into the liquid fraction as xylose and XOOs during HP is important, glucan release to glucose and gluco-oligomers (GOOs) must also be examined to obtain the whole picture of the pretreatment performance and to take advantage of the resulting fractions, both solid and liquid.

Xylan and glucan releases are usually reported as the results of HP conducted in batch-operation mode at the laboratory scale. For instance, Sidiras et al. [11], Ruiz et al. [12], Silva-Fernandes et al. [13], and Rossberg et al. [14] addressed polysaccharide release from WS as a function of solid loading, residence time, and temperature in batch reactors ranging from 100 mL to 3.75 L. Although isothermal conditions are reached at the reaction stage of HP, the heating and cooling rates are relatively slow. Usually, the heating time from the initial temperature to the desired reaction temperature varies from 15 to 80 min resulting in heating rates ranging from 2 to 7.1 °C/min. In addition, the cooling rates for laboratory-scale batch HP have rarely been described. For instance, Sidiras et al. [11] reported a cooling stage pretreatment temperature decreasing from 160 °C to room temperature in 30 min. Moreover, polysaccharide release during these long periods of heating and cooling is not distinguished from the corresponding release achieved in isothermal conditions. This might be the cause of significant discrepancies in the results of laboratory-scale HP available in the literature. For instance, Ruiz et al. [12] reported 39% as a maximum value for xylan release at 180 °C in a 0.6 L stirred batch reactor, while Sidiras et al. [11] reached 76% using a 3.7 L stirred batch reactor. Therefore, data related to the quantities of xylan release under HP regimes provided in the literature are not consistent.

Another laboratory-scale HP result with a high degree of uncertainty is the reaction time or residence time. According to Prunescu et al. [15], Rodriguez et al. [16], and Makishima et al. [17], shorter residence times than those reported for batch laboratory-scale HP studies are required to achieve similar amounts of released polysaccharides for HP conducted at the industrial or semi-pilot scale in continuous operation. The uncertainty in the residence time could imply significant readjustments in the biomass residence time systems, issues that Sieves and Stickel [18] and Jaramillo and Sanchez [19] reported as a technological challenge due to the rheological properties of the biomass. This might also represent a clear scaling-up problem for HP, as recently reported by Yue et al. [20]. Therefore, the results obtained at laboratory-scale HP are not trustworthy if needed for retrofitting or scaling-up purposes of HP at larger scales. This is because the continuous operation of an HP at commercial scales is governed by an isothermal regime. Heating is usually provided by saturated steam, while cooling is almost instantaneously achieved after steam explosion.

The effects of heating and cooling times in HP have been previously analyzed using the severity factor (SF) and modified severity factor (MSF) [21]. However, Ilanidis et al. [22] and Yu et al. [23] reported significant differences in terms of xylan release among pretreatments performed at the same SF with 1 L and 5000 L capacity reactors. Additionally, Conrad et al. [24] reported an SF interval (between four and five) in which linear proportionality with respect to hemicellulose conversion was not observed in 30 mL reactors. Moreover, no experimental validation of the MSF for short heating periods has been reported. Instead, the heating time or heating rates is commonly considered as the key operating condition in thermochemical LCB pretreatments [25,26].

In this contribution, experimental results expressed as polysaccharides release or, more specifically, xylan and glucan releases were obtained from a laboratory-scale batch HP of WS carried out under an overall isothermal operation. The isothermal operation was possible because both the heating and the cooling periods were very short or the heating and cooling rates were fast. Consequently, polysaccharides release can be considered negligible during both the heating and the cooling stages. The fast heating rate was induced by a fluidized sand bath, while fast cooling was achieved by the immersion of the batch HP reactor into a water bath. The main difference between the results described in this contribution and those already available is that they were obtained in rigorous isothermal conditions with a precise characterization of temperature trajectories in terms of heating and cooling rates and periods, as well as reaction times. Therefore, the results presented here could be considered as a basis for scaling-up or retrofitting HP calculations at commercial scales oriented to XOOs and GOOs production.

## 2. Results and Discussion

### 2.1. Characterization of Temperature Behavior for HP Experiments

The set of reaction temperatures and times, and heating and cooling times/rates for the HP experiments are compiled in Table 1. Temperature profiles for the longest reaction times are shown in Figure 1a.

The mean heating times ranged from 3 min in the HP experiments conducted at 150 and 160 °C to 6 min at higher temperatures. As shown in Figure 1b, isothermal reaction temperatures were reached after 3 min from 100 °C.

Furthermore, the mean heating rates were between 11 and 43 °C/min. Heating using a sand bath was compared to other techniques such as the steam chamber. For instance, Trajano et al. [27] reported a thermal study that revealed that the steam chamber offers better heat transfer performance compared to a sand bath. Nevertheless, these conclusions were obtained for a micro-scale reactor (10 mL). In addition, Shi et al. [28] concluded that the heating-up and cooling-down times for HP carried out by microwaves remain very close to those obtained for HP heated with a sand bath.

Regarding the cooling stage, the obtained mean cooling times ranged from 3 to 5 min with mean cooling rates from −15 to −40 °C/min. Although the cooling stage in large-scale HP in continuous operation is almost instantaneous, the cooling-down strategy of this work is three times faster than those previously described [11].

### 2.2. Analysis of pH Values in the HP Experiments

The dynamic evolution of pH at different reaction times is shown in Figure 1c. Generally, pH decreases when reaction temperature increases and remains bounded within 3.3 and 5.6. For reaction temperatures of 150 and 160 °C, the pH was between 4.6 and 5.6, an interval in which toxic inhibitors, such as furfural, and 5-hydroxymethylfurfural are not produced as a consequence of the poor production of monomeric sugars from soluble oligosaccharides [3]. The absence of furfural and HMF was confirmed with HPLC analysis. In addition, pH values at 180 to 220 °C showed similar patterns to those reported in reactions carried out in 3.7 L Parr reactors [5,11]. Additionally, a relation of −0.018 ± 0.003 pH units/°C was found when considering the isothermal reaction stage between 150 and 220 °C.

### 2.3. Analysis and Discussion of Xylan and Glucan Release Results

Figure 2 shows the xylan and glucan releases as monomers (%X_mr_ and %G_mr_, respectively), whereas Figure 3 describes the corresponding values of the xylan and glucan releases as monomers and oligomers (%X_mor_ and %G_mor_, respectively.) Numerical data are shown in Table 2.

Figure 3 shows both the weight/weight percentage and grams per liter of the reaction products to enable comparison with data already available in the literature. Notice that %X_mr_ and %G_mr_ were negligible in all the HP experiments conducted at 150 and 160 °C. Additionally, %X_mor_ and %G_mor_ for the HP experiments at *t* = 0 conducted at 150 °C were below 0.20 g/L. This amount is ten times lower than the value reported by Carvalheiro et al. [29] under similar reaction conditions.

The %X_mor_ and %G_mor_ were 6.0 (0.70 g/L) and 4.1 (0.81 g/L), respectively, at 150 °C after 60 min. Surprisingly, %X_mor_ exhibited a significant increase to 15%, or equivalently 1.7 g/L, when increasing the reaction temperature by 10 °C, whereas the increase in %G_mor_ was barely perceptible. In addition, a decrease in %G_mor_ at 150 and 160 °C was observed before the end of the reaction (i.e., at 60 min). This might represent the degradation of glucan released from highly reactive glucans. By comparison, this decrease was not observed for xylan because the rate of xylan released at 150 and 160 °C was greater than the production of degradants.

For higher temperatures (i.e., from 180 to 220 °C) at *t* = 0, changes in the %G_mor_ were proportional to changes in the reaction temperature. In contrast, monotonous decreases in the %X_mor_ and %G_mor_ were registered at 220 °C, suggesting a relatively high degradation rate of both polymers regarding the polymers released. These results are similar to those of Carvalheiro et al. [29] and Silva-Fernandes et al. [13], who reported degradation above 220 °C.

HP exhibited the largest value of %X_mor_ at 180 °C and 20 min. This value was 4.8 g/L, equivalent to %X_mor_ = 43, which is less than 2% of %X_mr_ according to the findings depicted in Figure 3. Similar operating conditions (i.e., 180 °C and 18 min) were reported by Rodriguez et al. [16] for HP experiments carried out at a semi-pilot scale in a continuous tubular reactor for diverse LCB feedstock. In that contribution, short heating times at the input of the reactor were observed and, due to steam explosion at the output, short cooling times were also reported, as in the present contribution. In addition, Rodriguez et al. [16] observed an evident decrease in %X_mor_ at 180 °C and 40 min, which was also observed in this study as a clear indicator of degradation products.

Th maximum value of %X_mor_ at 200 °C was 28 (or 3.1 g/L) with a reaction time of 10 min. This sample recorded the highest %X_mr_ (13), providing low glucan degradation. In addition, the maximum value of %X_mor_ at 200 °C reported here (i.e., 28 g/L) is higher than that obtained by Silva-Fernandes et al. [13], which is close to 20 g/L, and that was obtained at 210 °C in a 0.6 L stirred batch reactor with a heating rate equal to 6.2 °C/min, which is five- lower than the heating rate that is reported in this contribution (see Table 1). Additionally, a %X_mor_ decrease was observed from 3.1 to 1.8 g/L after 10 min, which is a symptom of a higher rate of degradant production.

Conducting a deeper analysis of the %G_mor_ and %G_mr_ results, the following findings were registered: First, %G_mor_ in the HP experiments conducted at *t* = 0 for samples pretreated at 160 to 220 °C was between 0.80 and 1.2 g/L. Additionally, the maximum value of %G_mor_ (1.2 g/L, which is equivalent to 5.5%) was observed in the experiments at 160 and 180 °C conducted for 30 min. This maximum value is only 0.2% higher than that obtained in the HP experiments at *t* = 0 conducted at 160 °C. In addition, similar concentrations of %G_mor_ at 200 and 60 min; and at 220 °C and 15, 20, and 30 min could be due to the highly reactive cellulose. %G_mor_ is lower than %X_mor_ because, as stated in the Introduction, this is a common behavior observed in HP because it favors xylan release. The results of %G_mr_ span from 0% to 2.7% (from 0 to 0.54 g/L). These relatively low values confirm that HP does not hydrolyze glucan. Furthermore, the %G_mr_ values do not show increases proportional to temperature as in other pretreatments, such as acid-catalyzed, that have been previously reported to favor glucan hydrolysis. In this contribution, the resulting glucan monomers may be produced from superficial, fast-reactive glucans on WS.

The discrepancies between the different polymer releases described in this contribution and those reported previously might be due to the fast heating-up and cooling-down periods of the experiments that do not contribute to polymer release. Although the success of hydrothermal pretreatment does not exclusively involve the release of xylan, XOOs has a potential market niche, and a good correlation with enzymatic saccharification has also been observed [30,31]. Furthermore, to have HP operating at rigorous isothermal conditions is fundamental to reach hydrogen and ethanol co-production via dark fermentation [32] and other eco-friendly fuel precursors [33].

## 3. Materials and Methods

### 3.1. Wheat Straw Characterization

Furrow-irrigated WS (*Triticum aestivum * L.; cultivar. Urbina S2007) was kindly supplied by a private agribusiness from Tepatitlán, Jalisco, Mexico (20°49′0.91″ N, 102°45′48.49″ W) with the following initial composition: glucan (41.32%) and xylan (19.54%). Compositional results were obtained according to the NREL laboratory analytical procedures [34]. The WS was harvested using conventional WS harvesting equipment and subsequently milled with an agricultural hammer mill (Azteca, Guadalajara Jal., México) equipped with a 1/2” sieve. WS particles of 14–18 Tyler mesh (1.41 mm to 1.00 mm) were collected with a Test-Master Testing Screen (Gilson, Lewis Center OH, USA). This size distribution was chosen to maximize sugar recovery (Rojas-Rejon and Sanchez [35]).

### 3.2. Experimental HP Runs

The HP experiments were carried out with WS samples (8 g) mixed with water (150 mL) in a tailor-made batch reactor of carbon steel SCG-80 with a 2in diameter and 6in length. The reactor was not provided with a stirring system to reproduce the poor mixing conditions described in continuous large-scale HP operations [18].

Reactor temperatures during the heating-up, reaction, and cooling-down stages of the HP experiments were measured through a thermowell with a Pt-100 RTD and recorded with I-7015 and I-7188E2 acquisition modules coupled to a data logger (ICPDAS, Shangai, China). Pressure within the reactor was measured with a 0–4 MPa ± 2.00% manometer (Instrutek, Guadalajara Jal., Mexico). The relationship between the temperature and the pressure was continuously monitored for temperature data validation.

Experimental runs started at 20 °C, pH = 5.6, and heating was provided by a TBSL12 fluidized sand bath (Accurate Thermal Systems, Hainesport, NJ, USA) equipped with automatic temperature control.

Although positive effects were reported when using several heating steps [36], a fast heating-up strategy was used to emulate large-scale HP operating conditions where the biomass is instantly in contact with steam at high temperatures.

The following temperature set points were evaluated during the reaction stage of the HP: 150, 160, 180, 200, and 220 °C. This interval of temperatures was previously identified to achieve high monomer yields and hemicellulose recovery in the HP of WS [29]. Reaction times between 0 and 60 min were registered after the reactor temperature reached the desired set point. Experiments labelled “*t* = 0” are those that were driven to the desired temperature set point, and then they were immediately cooled down. Once the reaction time finished, the reactor was removed from the sand bath and cooled down in a water bath to 100 °C, which was considered the final HP temperature. A total of 72 experiments were carried out, including repetitions of the different reaction temperature and reaction time combinations that are shown in Table 1.

### 3.3. Analysis on Xylan and Glucan Release

The strategy to quantify and report xylan and glucan release is based on the general reaction pathway of the hydrothermal pretreatment shown in Figure 4.

This pathway assumes that the masses of xylan ($m_{Xylan}^{0}$) and glucan ($m_{Glucan}^{0})$ in the WS at the beginning of the HP experiments are known and that the solid and liquid fraction produced after the HP can be recovered and analyzed qualitatively and quantitatively, respectively. One assumption that supports this pathway is that both unreacted xylan and glucan conform to the solid fraction. Another assumption is that monosaccharhplides (i.e., xylose and glucose), XOOs, furfural, and HMF are the main components of the liquid fraction. This fraction was analyzed after room temperature was reached as follows: first, it was centrifuged with a Heraeus Pico 17 centrifuge (ThermoFisher, Waltham, MA, USA) at 13,000 rpm for 5 min; then it was filtrated with a 45 µm acrodisc syringe filter. In addition, the pH of the filtrate was measured using an Orion Star A211 benchtop pH meter (ThermoFisher, Waltham MA, USA).

The concentrations of the degradation compounds (i.e., furfural and HMF) in the liquid phase were measured using the 2695 HPLC (Waters Alliance, Milford MA, USA). The xylose and glucose concentrations in the liquid fraction were determined via a YSI 2700D biochemistryanalyzer (Mashall Scientific, Hampton, NH, USA). Then the mass of xylose $m_{Xylose}^{HT}$ and the mass of glucose $m_{Glucose}^{HT}$ were computed. The percentage of xylan (%X_mr_) and glucan (%G_mr_) release as monomers after HP were computed with the following equations:(1)$$\%X_{mr}=100\left(\frac{m_{Xylose}^{HT}}{m_{Xylan}^{0}}\frac{1}{FEX}\right)$$
(2)$$\%G_{mr}=100\left(\frac{m_{Glucose}^{HT}}{m_{Glucan}^{0}}\frac{1}{FEG}\right)$$
where FEX = 1.13 and FEG = 1.11 are the stoichiometric factors between the monomers and oligomers of xylan and glucan, respectively.

In addition, to quantify the released oligomers present in the liquid fraction that was hydrolyzed in the presence of H_2_SO_4_ (4%) at 121 °C for 30 min. Later, the xylose and glucose concentrations were determined using a YSI 2700D biochemistryanalyzer (Mashall Scientific, Hampton NH, USA). that was applied to the after-hydrolysis (AH) liquid. Then, the mass of xylose $\left(m_{Xylose}^{AH}\right)$ and the mass of glucose $\left(m_{Glucose}^{AH}\right)$ were computed. The masses of oligomers could be calculated with the following expressions:(3)$$m_{XyloO}=m_{Xylose}^{AH}-m_{Xylose}^{HT}$$
(4)$$m_{GlucoO}=m_{Glucose}^{AH}-m_{Glucose}^{HT}$$

Finally, the polymers released from xylan to both xylose and XOOs (%X_mor_) and from glucan to both glucose and GOOs (%G_mor_) after HP were computed as follows:(5)$$\%X_{mor}=100\left(\frac{m_{XyloO}+m_{Xylose}^{HT}}{m_{Xylan}^{0}}\frac{1}{FEX}\right)=100\left(\frac{m_{Xylose}^{AH}}{m_{Xylan}^{0}}\frac{1}{FEX}\right)$$
(6)$$\%G_{mor}=100\left(\frac{m_{GlucoO}+m_{Glucose}^{HT}}{m_{Glucan}^{0}}\frac{1}{FEG}\right)=100\left(\frac{m_{Glucose}^{HT}}{m_{Glucan}^{0}}\frac{1}{FEG}\right)$$

## 4. Conclusions

Laboratory-scale batch HP experiments of WS were conducted under an overall isothermal regime since the heating and cooling stages were carried out in very short periods with a minimum effect on the polymer release. The results of these experiments in terms of xylan and glucan release showed a clear difference with those already reported in the literature for HP at the laboratory scale but are similar to those reported for HP conducted at commercial scales in continuous and isothermal operation.

## Figures and Tables

**Figure 1 molecules-27-00026-f001:**
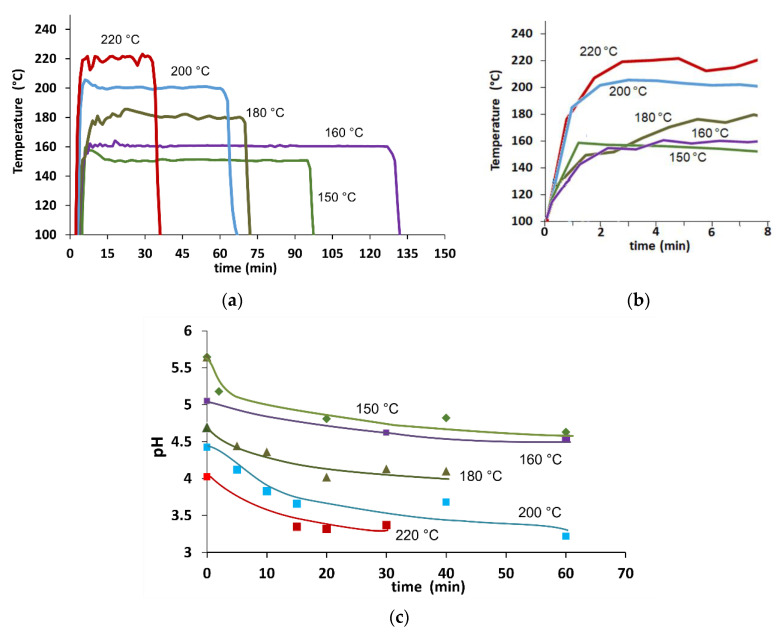
(**a**) Typical behavior of reaction temperature and (**b**) heating ramps in HP experiments of WS at the longest reaction times. (**c**) Mean pH profiles in HP experiments of WS as a function of time. The continuous lines are the tendencies of the experimental pH values.

**Figure 2 molecules-27-00026-f002:**
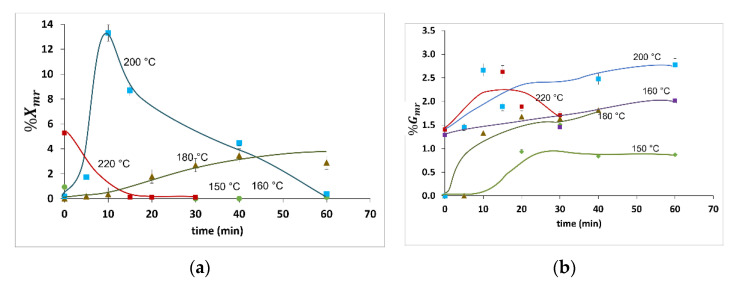
Polymers release to monomers as a function of isothermal reaction time and temperature of (**a**) xylose (%X_mr_) and (**b**) glucose (%G_mr_). Data points represent the averages of 3 experiments. Error bars represent standard uncertainties.

**Figure 3 molecules-27-00026-f003:**
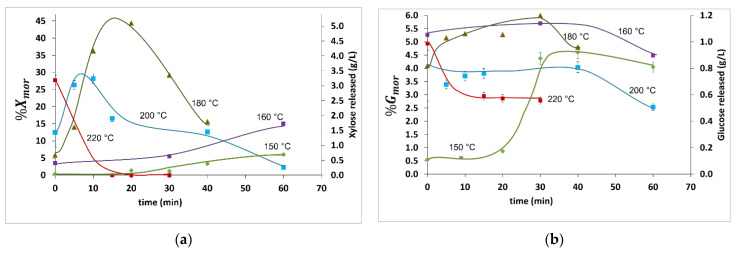
Polymers release to monomers and oligomers of (**a**) xylose (%X_mor_) and (**b**) glucose (%G_mor_). Data points represent the averages of 3 experiments. Error bars represent standard uncertainties.

**Figure 4 molecules-27-00026-f004:**
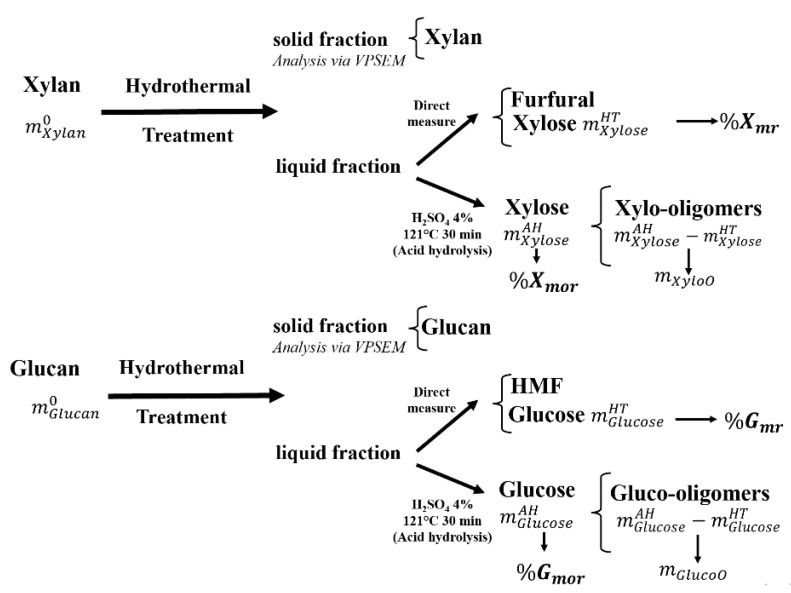
Xylan and glucan release pathways. %X_mr_ and %G_mr_ are the monomers released from xylan to xylose and from glucan to glucose, respectively, while %X_mor_ and %G_mor_ are the polymers released from xylan to both xylose and XOOs and from glucan to both glucose and GOOs, respectively.

**Table 1 molecules-27-00026-t001:** Characterization of reaction temperature trajectories.

Reaction Temperatures (°C)	Reaction Times (min)	Mean Heating Times (min), Rates (°C/min)	Mean Cooling Times (min), Rates (°C/min)
150	0, 20, 30, 40, 60	≈3, 43	≈3, −17
160	0, 30, 60	≈5, 28	≈4, −15
180	0, 5, 10, 20, 30, 40	≈6, 27	≈4, −20
200	0, 5, 10, 15, 40, 60	≈6, 30	≈3, −33
220	0, 15, 20, 30	≈6, 34	≈3, −40

**Table 2 molecules-27-00026-t002:** Results of laboratory-scale batch HP experiments.

Reaction Temperature (°C)	Reaction Time (min)	%X_mr_	%G_mr_	$\%X_{mor}/$ $Xylan\;Release\left(\frac{g}{L}\right)$	$\;\;\;\%G_{mor}/$ $Glucan\;Release\left(\frac{g}{L}\right)$	pH
150	0	0.010	0.01	0.40/0.00	0.60/0.10	5.3
150	20	0.020	0.93	1.4/0.20	0.90/0.20	4.8
150	30	0.030	1.5	1.2/0.10	5.3/1.1	4.8
150	40	0.030	0.84	3.3/0.40	4.6/0.90	4.8
150	60	0.13	0.87	6.0/0.70	4.1/0.81	4.6
160	0	0.010	1.3	3.5/0.40	5.3/1.1	5.1
160	30	0.020	1.5	5.5/0.60	5.5/1.2	4.4
160	60	0.080	2.0	15/1.7	4.5/0.90	4.5
180	0	0.010	0.010	5.9/0.70	4.1/0.80	4.7
180	5	0.19	0.010	14/1.6	5.2/1.0	4.4
180	10	0.34	1.3	36/4.0	5.3/1.1	4.4
180	20	1.8	1.7	43/4.8	5.3/1.1	4.0
180	30	2.7	1.6	29/3.3	5.5/1.2	4.1
180	40	3.5	1.8	16/1.7	4.8/1.0	4.1
200	0	0.19	0.010	13/1.4	4.1/0.80	4.4
200	5	1.7	1.5	26/2.9	3.4/0.70	4.1
200	10	13	2.7	28/3.1	3.7/0.70	3.8
200	15	8.0	1.9	17/1.8	3.8/0.80	3.7
200	40	4.5	2.5	13/1.4	4.0/0.80	3.8
200	60	0.37	2.8	2.3/0.3	2.5/0.50	3.5
220	0	5.3	1.4	28/3.1	4.9/1.0	4.0
220	15	0.12	2.6	0.20/0.00	3.0/0.60	3.4
220	20	0.090	1.9	0.20/0.00	2.9/0.60	3.3
220	30	0.11	1.7	0.00/0.00	2.8/0.60	3.4

## Data Availability

The data presented in this study are available on request from the corresponding author.

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
