# Peer review of "Polysaccharides Release in a Laboratory-Scale Batch Hydrothermal Pretreatment of Wheat Straw under Rigorous Isothermal Operation"

_molecules, 2021, doi:10.3390/molecules27010026_

Round 1

Reviewer 1 Report

The paper is focused with the hydrothermal pretreatment of wheat straw under isothermal conditions. Comparing with previous publications on the subject the authors attempted to explain the variation in reported glucan/xylan releases that can be found in the literature. Although, the novelty of the paper is limited but it has certain utility to those interested in wheat straw as raw material in biorafineries. The paper is well organized but it required some further editing and clarification of some aspects.

l.23 - 43.28% - too many significant digits, 43.3 or 43% is sufficient.

l.30 – 980 million – of dry mass?

l.68 – is the meaning of this sentence = the data in the literature is not consistent on the quantities of xylan release under HP regimes?

l.135 – figure 1c

Fig 2 and fig 3 – uniform style/size of the figures would be better

Fig 2 and fig 3 heading – instead “average values and …” use “data points represents averages of n=? experiments. Error bars represents standard deviations.

In the latter case it would be better to use standard uncertainties (that is standard deviations of means, not standard deviations)

l.160 – the amount is near 10% -what is the actual meaning here? – “in comparison with… ten times lower amount was observed under…”?

l.177 – 43.28% - as in l.23

l.214 – as above

l.267 – Waters

missing or redundant spaces between values and units should be corrected in the text

Author Response

The authors appreciate the comments provided by the reviewer. All the comments are duly addressed  as follows:

l.23 - 43.28% - too many significant digits, 43.3 or 43% is sufficient.

R.- The reviewer is right. Numerical results were reported with different significant digits. In the new version of the manuscript, we reported all the numerical results only with two significant digits.

l.30 – 980 million – of dry mass?

R.- Definitively, establishing whether is wet or dry mass is extremely important. However, the source of information does not declare the condition (see the following paragraph of reference [1]):

As the type of mass (wet or dry) of available wheat straw is not provided in the source or other bibliographical sources, the authors decided to reproduce the information provided in the cited reference without further comments.

l.68 – is the meaning of this sentence = the data in the literature is not consistent on the quantities of xylan release under HP regimes?

R.- Yes, this is correct and if there is no problem, we would like to replace the original sentence with the sentence proposed by the reviewer as follows:

Therefore, data related to the quantities of xylan release under HP regimes provided in the literature are not consistent.

l.135 – figure 1c

R.- Yes, thank you very much to help us to improve the paper by pointing out these mistakes.

Fig 2 and fig 3 – uniform style/size of the figures would be better

R.- The reviewer is right. Figure 2 and Figure 3 differ in format because the latter contains an additional y-axis on the right side of the figure denoting xylan and glucan release as monomers and oligomers in g/L (xylose and glucose released). However, we included such additional y-axis only in Figure 3 because xylose and glucose released in g/L are the units to compare our results with those previously reported.

If the editor agrees, we would like to maintain Figure 2 as in the original version of the paper. This Figure describes released  xylan and glucan release as monomers in percentage (i.e. and ) which are not usually reported in HP experiments but unequivocally establish the effectiveness  of the pretreatment, as opposed to the use ofg/L values. In this latter case, initial concentrations must be known to compare the effectiveness of the pretreatment being considered.

Regarding the size of the figures, the reviewer is right again. To make the size uniform we will provide the source files of all figures to the editor.

Finally, we have corrected the typo on the right y-axis of Figure3b. We have replaced glucosa released with glucose released.

Fig 2 and fig 3 heading – instead “average values and …” use “data points represents averages of n=? experiments. Error bars represents standard deviations.

In the latter case it would be better to use standard uncertainties (that is standard deviations of means, not standard deviations)

R.- This is an excellent recommendation from the reviewer. We have replaced the phrase “average values” for “data points represent averages of 3 experiments. Error bars represent mean standard uncertainties” in Figure 2 and Figure 3.

l.160 – the amount is near 10% -what is the actual meaning here? – “in comparison with… ten times lower amount was observed under…”?

R.- The meaning here is that the obtained quantity 0.2g/L is ten times lower than the amount observed by Carvalheiro. This has been corrected as

This amount is ten times lower than the value reported by Carvalheiro et al. [29] under similar reaction conditions.

l.177 – 43.28% - as in l.23

R.- All amounts in the paper related to xylose and glucose releases have been adjusted to two significant digits.

l.214 – as above

R.- All the amounts in the paper related to xylose and glucose releases have been adjusted two significant digits

l.267 – Waters

R.- We agree with the reviewer, the brand of the HPLC is Waters, not Water. This typo was corrected.

missing or redundant spaces between values and units should be corrected in the text

 R.-. We have used the show/hide button in Word to detect and correct redundant spaces between values and units.

Reviewer 2 Report

This paper analyzed the sugar generation by the hydrothermal treatment of wheat straw according to the reaction conditions. However, I think some corrections are needed to be published.

1. In the case of Figure 1c, the two conditions (pH when the reaction starts, reaction temperature) are different between the reaction conditions of HP. Therefore, it is difficult to intuitively compare the effect of reaction conditions on pH. Even, Figure 1c is not explained in the text, so it is more difficult to understand this part.

2. In Figure 2b, the trend of glucose yield according to the reaction conditions is somewhat different from the generally known trend. As the reaction temperature increased from 150 °C to 160 °C, the glucose yield increased, but at 180 °C, it decreased again. However, when the reaction temperature was raised to 200 °C again, the glucose yield was greatly increased, and at 220 °C it was lowered again. I think that further explanation of the tendency of this change in glucose yield is necessary.

3. In '2.2 Analysis of pH values in the HP experiments', it was said that the production of furfural and 5-HMF was inhibited when the pH was 3.3~5.6. What is the reason? References are indicated, but a brief explanation is needed in the text for better understanding.

Author Response

The authors appreciate the comments provided by the reviewer. All the comments are duly addressed in the attached file R2.pdf

Reviewer 3 Report

The manuscript titled " Polysaccharides release in a lab-scale batch hydrothermal pretreatment of wheat straw under rigorous isothermal operation" is over all all well written and interesting. The experimental design is appropriate and techniques are well chosen. I recommend publication of the manuscript after authors take care of the typos. 

Author Response

The authors thank the comments provided by the reviewer. Typos were corrected.